# Hierarchical Time-Aware Mixture of Experts for Multi-Modal Sequential Recommendation

## Abstract

Multi-modal sequential recommendation (SR) leverages multi-modal data to learn more comprehensive item features and user preferences than traditional SR methods, which has become a critical topic in both academia and industry. Existing methods typically focus on enhancing multi-modal information utility through adaptive modality fusion to capture the evolving of user preference from user-item interaction sequences. However, most of them overlook the interference caused by redundant interest-irrelevant information contained in rich multi-modal data. Additionally, they primarily rely on implicit temporal information based solely on chronological ordering, neglecting explicit temporal signals that could more effectively represent dynamic user interest over time. To address these limitations, we propose a **H**ierarchical time-aware **M**ixture of experts for multi-modal **S**equential **R**ecommendation (HM4SR) with a two-level Mixture of Experts (MoE) and a multi-task learning strategy. Specifically, the first MoE, named Interactive MoE, extracts essential user interest-related information from the multi-modal data of each item. Then, the second MoE, termed Temporal MoE, captures user dynamic interests by introducing explicit temporal embeddings from timestamps in modality encoding. To further address data sparsity, we propose three auxiliary supervision tasks: sequence-level category prediction (CP) for item feature understanding, contrastive learning on ID (IDCL) to align sequence context with user interests, and placeholder contrastive learning (PCL) to integrate temporal information with modalities for dynamic interest modeling. Extensive experiments on four public datasets verify the effectiveness of HM4SR compared to several state-of-the-art approaches. Our code is available at https://anonymous.4open.science/r/HM4SR-8DD7/.

## CCS Concepts

• **Information systems → Recommender systems**.

## Keywords

Sequential Recommendation, Multi-modal Recommendation, Mixture of Experts, Temporal Information

**ACM Reference Format:**
Anonymous Author(s). 2025. Hierarchical Time-Aware Mixture of Experts for Multi-Modal Sequential Recommendation. In *Proceedings of Proceedings of the ACM on Web Conference (WWW '25)*. ACM, New York, NY, USA, 11 pages. https://doi.org/XXXXXXX.XXXXXXX

## 1 Introduction

Sequential recommendation (SR) aims to predict the subsequent interactions of users by analyzing their historical behaviors in chronological order [18, 31]. Traditional SR methods rely on item-IDs solely to develop sequence encoders [13, 19, 21, 25, 43], and the abundant multi-modal item descriptions are ignored and unused, which may encapsulate potential user interests. Recently, the multi-modal SR task has been proposed to integrate various types of multi-modal data like text and images as semantic supplements for user-item interactions [1, 15, 16, 22, 32]. In this context, these approaches significantly enhance recommendation quality by capturing more fine-grained item features and better modeling user interests, thereby attracting increasing attention from both academia and industry.

In the literature, conventional multi-modal SR approaches primarily focus on modality fusion to improve recommendation accuracy. For instance, MV-RNN [4] conducts an initial modality fusion through addition, concatenation, and reconstruction, while UniS-Rec [15] applies Mixture of Experts to transfer semantics from text representations to ID embeddings. Moving beyond direct fusion, many studies try to explore adaptive fusion for more flexible multi-modal modeling. For example, MISSRec [30] employs a lightweight fusion module to learn user dynamic attention on different modalities. Next, MMSR [16] reveals that fusing modalities at different stages impacts model sensitivity to interest patterns and proposes adaptive modality fusion based on heterogeneous graph neural networks. TedRec [35] applies Fast Fourier Transform to process embedding sequences of ID and text for sequence-level semantic fusion from the frequency domain. Despite these advancements, most of them overlook the hindrance of multi-modal redundant information on the learning of true user interests. They also model multi-modal sequences by solely leveraging implicit temporal information contained in the chronological order of behaviors, neglecting explicit temporal data that indicates user dynamic interest changes, such as timestamps and time intervals.

Although considerable progress has been made, there are still some critical challenges for multi-modal SR: **(1)** While the richness of multi-modal information enhances item and user representation learning, its redundant interest-irrelevant parts may complicate the extraction of key information that reflects user true preferences from multi-modal data. For example in Figure 1, the text for the fourth item includes usage directions, and the image contains details about the person on the package, both are irrelevant to user interest. **(2)** Explicit temporal information is dynamic and diverse, making it challenging and potentially unstable to leverage for modeling evolving user interests in multi-modal sequence encoding. In Figure 1, the user has been interested in the "Crusader" brand for a long time but recently preferred summer-related products like sunscreens, indicating the complexity of using explicit temporal data to capture dynamic interests. **(3)** Sequence data sparsity causes information

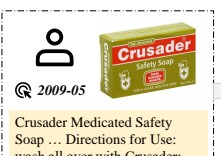 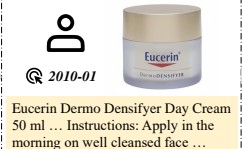 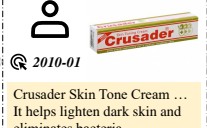 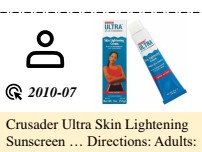 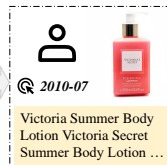 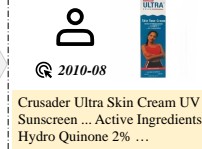

**Figure 1: A toy example of user-item interaction sequence.**

loss for user interest modeling, necessitating multi-faceted auxiliary supervised signals to replenish interest-related information. However, due to the complexity of multi-modal data, how to get informative signals presents a challenge. In Figure 1, the model has to learn user interest from a short interaction sequence, and items like "Crusader Skin Tone Cream" also have limited interactions, highlighting the sparsity in both user and item data.

To address aforementioned challenges, we propose a novel method named **H**ierarchical Time-Aware **M**ixture of Experts for **M**ulti-Modal **S**equential **R**ecommendation (HM4SR). Specifically, first for initial item representations, we use BERT [6] and ViT [7] to extract textual and visual information, and maintain an embedding matrix for item-IDs. Next, we develop a two-level hierarchical Mixture of Experts (MoE). The first level, namely Interactive MoE, applies specialized experts to process all modalities of an item concurrently for the extraction of key information relevant to user interests. The second level, termed Temporal MoE, encodes timestamps into embeddings of intervals and absolute time to model explicit temporal information. It then utilizes these embeddings to select experts to process item features, thereby integrating explicit temporal information into multi-modal learning. To capture user dynamic interests, HM4SR employs Transformers [29] to compute embedding sequences of each modality. Furthermore, we design three auxiliary supervision tasks to provide multi-faceted and informative training signals. We first devise a sequence-level category prediction (CP) task to fuse category information. Next, contrastive learning on ID (IDCL) improves alignments between sequence representations and ground truth embeddings on ID, enhancing user preference learning. To strengthen modality and dynamic temporal information integration, placeholder contrastive learning (PCL) promotes consistency between original sequences and their augmented sequences with time-based placeholders. Overall, we summarize our contributions as follows:

- In this paper, we propose a novel HM4SR method for multi-modal SR. To the best of our knowledge, we are the first to leverage explicit temporal information to enhance multi-modal user interest learning.
- We design a hierarchical MoE structure. At the first level, the Interactive MoE extracts key information related to user interests from rich multi-modal data. At the second level, the Temporal MoE incorporates dynamic explicit temporal information into the modality modeling process.
- To provide multi-faceted and effective training signals, we develop a multi-task learning strategy including three auxiliary supervision tasks, i.e., CP, IDCL, and PCL.
- Comprehensive experiments on four real-world datasets verify the effectiveness of our proposed HM4SR.

## 2 Related Work

In this section, we provide previous research relevant to our work, including traditional and multi-modal sequential recommendation.

### 2.1 Traditional Sequential Recommendation

Sequential recommendation targets to predict the next item users may be interested in based on their behavior sequences. Early works focus on different sequential encoding models to process user behavior sequences. For instance, GRU4Rec [13], Caser [27], SASRec [19] and BERT4Rec [26] utilize RNN, CNN, self-attention and BERT structures as their basic encoders, respectively. Since time information apparently represents the changes and transitions of user interest, many studies try to enhance SR methods with time-aware designs. For example, TiSASRec [20] exploits time intervals for self-attention-based models to capture evolving information in behavior sequences. MEANTIME [3] exploits multiple types of time embeddings to improve sequential interest modeling. TGCL4SR [39] devises temporal graph contrastive learning with timestamp perturbation augmentation to enhance item temporal transition pattern encoding. Furthermore, FEARec [9] designs a frequency rump structure, and hybrids the time domain self-attention encoder with the frequency domain self-attention module to grasp both low-frequency and high-frequency patterns. TiCoSeRec [5] utilizes five time-aware sequence-level augmentation operations to unify the time distribution of user behaviors and conduct contrastive learning. However, the above methods model evolving behavior sequences without exploring multi-modal data, and thus cannot fully exploit multi-modal features which reflect user interests.

### 2.2 Mutli-Modal Sequential Recommendation

Multi-modal SR has emerged to leverage multi-modal information of items to capture user interests, which significantly enhances recommendation quality [17, 22, 37]. Existing studies mainly concentrate on integrating modalities to augment recommendation accuracy. For example, MV-RNN [4] fuses modal data through addition, concatenation, and reconstruction. UniSRec [15] applies MoE to facilitate semantic transfer on text representations and add them to the ID embeddings. Successively, many works further design adaptive fusion modules to improve multi-modal fusion effectiveness. For instance, MISSRec [30] implements a lightweight fusion mechanism to discern user dynamic attention across modalities for adaptable fusion. MMSR [16] devises adaptive fusion with heterogeneous graph neural networks to flexibly utilize the interplay between modalities. M3SRec[1] uses modality-specific MoE and cross-modal MoE for multi-sided modality integration. In addition, TedRec [35] leverages Fast Fourier Transform to process the embedding sequences of IDs and textual contents for sequence-level

semantic fusion in the frequency domain. While most methods rely solely on implicit temporal information, they often neglect dynamic explicit temporal data, redundant modal filtering, and the extraction of interest-relevant information. In contrast, our proposed HM4SR addresses these challenges by explicitly incorporating dynamic temporal information and introducing auxiliary supervision tasks to enhance dynamic user interest modeling.

## 3 Problem Definition

Multi-modal sequential recommendation aims to exploit multi-modal item information and historical user behavior to make personalized recommendations for the next user interaction. Let us assume a set of users $\mathcal{U}$ and a set of items $\mathcal{V}$. Each item $v \in \mathcal{V}$ is represented as $v = \langle i_{id}, i_{txt}, i_{img} \rangle$, where $i_{id}$ denotes the item ID, while $i_{txt}$ and $i_{img}$ refer to the associated text and image content of $v$, respectively. We denote the historical behavior sequence $S$ of user $u \in \mathcal{U}$ as $S = \{v_1, v_2, \ldots, v_{|S|}\}$, where $v_i \in \mathcal{V}$ represents the interacted item at the $i$-th time step. The corresponding timestamp sequence is denoted as $T = \{t_1, t_2, \cdots, t_{|S|}\}$.

Given a user $u$ and $u$'s historical behavior sequence $S$ with multi-modal item information, the objective of multi-modal sequential recommendation is to predict an item $v \in \mathcal{V}$ that the user $u$ might be interested in at the $(|S| + 1)$-th time step. This can be formulated as finding the item $v$ that maximizes the conditional probability given the sequence of interactions:

$$v = \operatorname{argmax}_{v \in \mathcal{V}} P(v_{|S|+1} = v|S). \tag{1}$$

## 4 Methodology

In this section, we introduce the technical details of our proposed HM4SR. As illustrated in Figure 2, HM4SR consists of four main components: 1) *Initial Item Representation* module extracts the ID, text, and image features for items as the initial representations. 2) *Hierarchical Mixture of Experts* module designs two levels of MoE, named Interactive MoE and Temporal MoE, to mine key interest features from the multi-modal information interaction and harness explicit temporal information for multi-modal sequence modeling by inputting time embeddings to gating routers. 3) *User Interest Learning* module encodes user behavior sequences with different modalities to learn user preferences. The prediction is based on the sum of logits from three modalities. 4) *Multi-task Learning* strategy is designed to enhance main recommendation task training, including sequence-level category prediction and contrastive learning on ID. Placeholder contrastive learning is further proposed to deepen relationship learning between item features and time information.

## 4.1 Initial Item Representation

We obtain item initial representations from three modalities in real-world recommendation scenarios: ID, text, and images. For ID, we initialize an ID embedding matrix $M_{id} \in \mathbb{R}^{|\mathcal{V}| \times d}$, where $d$ denotes the size of the hidden dimension. We let $x_{id}$ represent the initial ID representation of the item $v \in \mathcal{V}$. As for text and image data, we encoder them with pre-trained models for better representation.

First, for text, we apply a widely used pre-trained BERT [6] to extract text features to capture user preference from textual semantics. Given the words $\{w_1, w_2, \ldots, w_L\}$ of $v$ in textual information of items, we first concatenate them with a special symbol $[CLS]_{txt}$

to form the input sentence. Since $[CLS]_{txt}$ can convey the semantics of the whole sentence, we use the embedding of $[CLS]_{txt}$ to represent the text features. Here, we input the combined sentence into pre-trained BERT to obtain the text feature as follows:

$$f_{txt} = \text{BERT}([[CLS]_{txt}; w_1; w_2; \ldots; w_L]), \tag{2}$$

where $f_{txt} \in \mathbb{R}^{d_{txt}}$ is the final hidden vector for $[CLS]_{txt}$, and $[;]$ denotes the concatenation operation.

Second, for images, we process the visual information by the pre-trained visual model ViT [7]. The image $w_{img}$ of item $v$ is divided into several patches $\{p_1, p_2, \ldots, p_N\}$, and then these patches are transformed into a sequence. Next, we input the patch sequence into ViT [7] as follows:

$$f_{img} = \text{ViT}([[CLS]_{img}; p_1; p_2; \ldots; p_N]), \tag{3}$$

where $f_{img} \in \mathbb{R}^{d_{img}}$ is the final hidden vector for $[CLS]_{img}$.

Moreover, to convert the dimensionality of text embeddings and image embeddings into the same dimension size as ID embeddings, we employ two respective linear layers to change their dimensions:

$$x_{txt} = W_{txt} f_{txt} + b_{txt}, \tag{4}$$

$$x_{img} = W_{img} f_{img} + b_{img}, \tag{5}$$

where $x_{txt}$ is the initial text representation of $v$, $x_{img}$ is the initial image representation of $v$, $W_{txt} \in \mathbb{R}^{d_{txt} \times d}$, $W_{img} \in \mathbb{R}^{d_{img} \times d}$, $b_{txt} \in \mathbb{R}^d$ and $b_{img} \in \mathbb{R}^d$ are trainable parameters, while $f_{txt}, f_{img}$ are frozen in the training process.

Meanwhile, order information is vital for sequence modeling in SR [1, 19], thus we add the corresponding position embedding $p \in \mathbb{R}^d$ on three embeddings:

$$e_m = x_m + p, \tag{6}$$

where $m \in \{id, txt, img\}$ denotes the modality.

## 4.2 Hierarchical Mixture of Experts

To model users' interests for recommendations from multi-model information, we design the two-level hierarchical Mixture of Experts including Interactive MoE and Temporal MoE. Interactive MoE extracts user interest-related item features to facilitate modality data learning, while Temporal MoE introduces explicit temporal information for dynamic interest modeling with the merit of specialization. In previous studies, the effect of MoE has been demonstrated in many recommendation scenarios, because it can augment the learning ability and flexibility of recommendation models by obtaining specialized item representations and user preferences from multiple aspects [2, 24, 34, 40]. MoE in multi-modal SR typically employs multiple expert networks to process modality semantic information and achieves adaptive combination by a gating network [15]. The following are details of our Interactive MoE and Temporal MoE.

*4.2.1 Interactive MoE.* The target of Interactive MoE is to enhance item key feature extraction and avoid redundant information of each modality. The MoE for multi-modal SR usually processes information from one modality [1, 15, 35], which lacks interactions between different modalities and thus hinges on more informative item feature learning. Different from them, we let each expert in Interactive MoE process all modality information of items and route experts based on a target modality to achieve effective modality

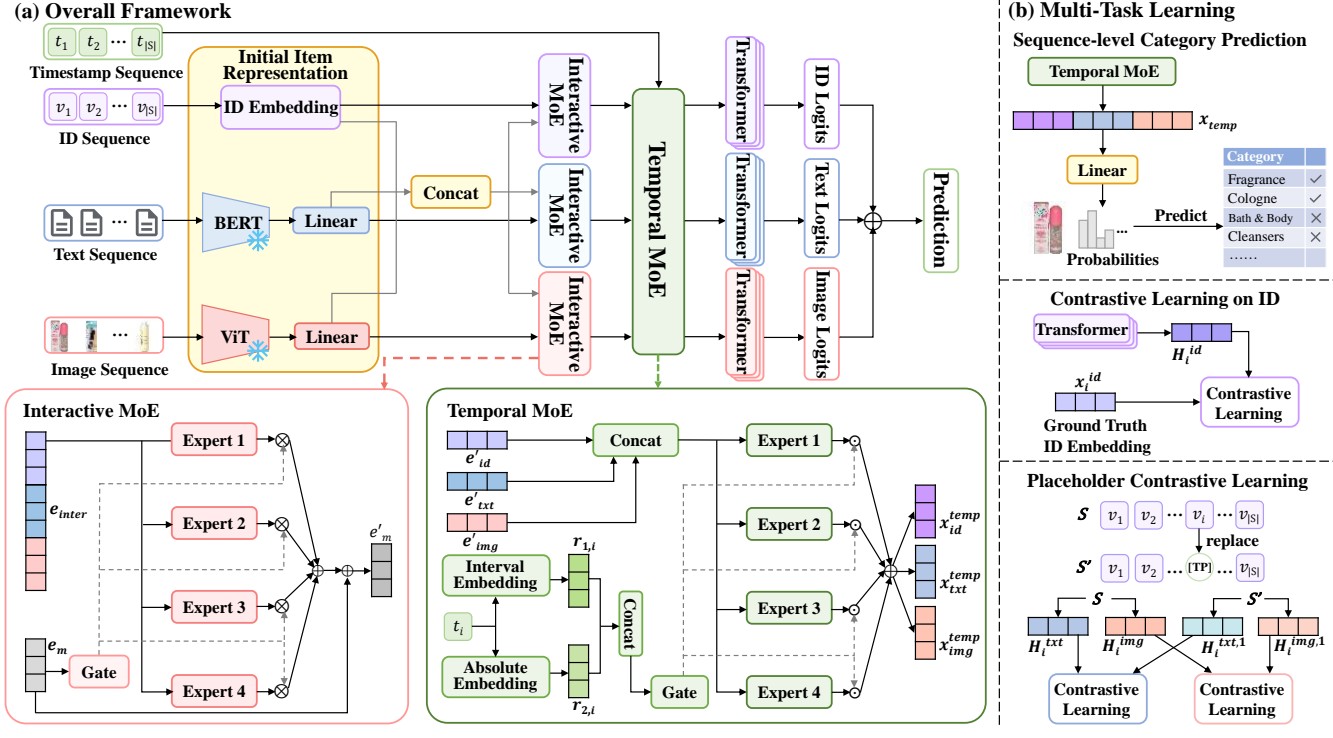

**Figure 2: The overall architecture of the proposed HM4SR.**

interaction learning. In specific, given the ID, text, and image representations of an item $e_{id}$, $e_{txt}$ and $e_{img}$ respectively, experts on the target modality $m \in \{id, txt, img\}$ process them as follows:

$$e_{inter} = \left[ e_{id}; e_{txt}; e_{img} \right], \tag{7}$$

$$x_m^{inter} = \alpha_m \sum_{i=1}^{k_1} g_i^m \cdot \left( W_{i,m}^{inter} e_{inter} + b_{i,m}^{inter} \right), \tag{8}$$

where $x_m^{inter}$ is the fused expert output, $\alpha_m$ is a trainable parameter to control modality interaction intensity on the modality $m$, $k_1$ denotes the number of experts in Interactive MoE, $W_{i,m}^{inter} \in \mathbb{R}^{3d \times d}$ and $b_{i,m}^{inter} \in \mathbb{R}^d$ are the learnable weight and bias of the $i$-th expert. $g_i^m$ represents the routing weight of the $i$-th expert on the modality $m$ from the routing vector $g^m$ by the following gating router:

$$g^m = \text{Softmax} \left( W_{1,m} e_m + b_{1,m} \right), \tag{9}$$

where $W_{1,m} \in \mathbb{R}^{d \times k_1}$ and $b_{1,m} \in \mathbb{R}^{k_1}$ are the learnable weight and bias of the gating routers. After obtaining $x_m^{inter}$, we take the residual connection to get the outputs of Interactive MoE:

$$e_m' = e_m + x_m^{inter}. \tag{10}$$

With the procedure above, item interest-related key information is semantically enhanced by modality interaction learning.

*4.2.2 Temporal MoE.* User interest usually changes dynamically and complicatedly, which can be more expressly indicated by explicit temporal information. Therefore, we further propose Temporal MoE as the second level to incorporate explicit temporal information into multi-modal learning, including intervals between user behaviors and absolute timestamps. To begin with, given the

timestamp sequence $T = \{t_1, t_2, \ldots, t_{|S|}\}$ of the sequence $S$, the corresponding time interval sequence is formulated as follows:

$$\left\{ a_1, a_2, \ldots, a_{|S|} \right\} = \left\{ 0, t_2 - t_1, \ldots, t_{|S|} - t_{|S|-1} \right\}, \tag{11}$$

where $a_i$ denotes the corresponding interval of the item $v_i \in S$. Considering that different time intervals indicate different user interest transition [36], we apply the following function to calculate the corresponding position of $a_i$:

$$\text{pos}_i = \lfloor \mu \log (a_i + 1) \rfloor, \tag{12}$$

where $\mu$ is a scaling parameter to control the total number of interval positions. We maintain an embedding matrix $M_t$ for intervals, and then we can get the interval embedding $r_{1,i} \in \mathbb{R}^d$ corresponding to $\text{pos}_i$. For absolute timestamps, inspired by the positional embedding in Transformer [29], we use the following function to obtain the time embedding of the timestamp $t$ for direct time information:

$$r_{2,i}^i = \cos \left( \frac{l_i t}{freq^{\frac{i}{d}}} + z_i \right), \tag{13}$$

where $r_{2,i}^i$ is the $i$-th value of $r_{2,i} \in \mathbb{R}^d$, $freq$ is an adjustable hyper-parameter, $l_i$ and $z_i$ are trainable parameters. After that, the temporal embeddings are inputted to the gating router of Temporal MoE to obtain the routing weight:

$$g' = \text{Softmax} \left( W_2 [r_{1,i}; r_{2,i}] + b_2 \right), \tag{14}$$

where $W_2 \in \mathbb{R}^{2d \times k_2}$ and $b_2 \in \mathbb{R}^{k_2}$ are the learnable weight and bias of the gating router respectively, $k_2$ is the number of experts. Then given the outputs from Interactive MoE $e_{id}'$, $e_{txt}'$ and $e_{img}'$ of

$v_i$, the output calculation of Temporal MoE is defined as below:

$$\boldsymbol{e}_{\text{temp}} = \left[\boldsymbol{e}'_{id}; \boldsymbol{e}'_{txt}; \boldsymbol{e}'_{img}\right], \tag{15}$$

$$\left[\boldsymbol{x}^{\text{temp}}_{id}; \boldsymbol{x}^{\text{temp}}_{txt}; \boldsymbol{x}^{\text{temp}}_{img}\right] = \boldsymbol{x}_{\text{temp}} = \sum_{i=1}^{k_2} g'_i \cdot \left(\boldsymbol{W}^{\text{temp}}_i \odot \boldsymbol{e}_{\text{temp}}\right), \tag{16}$$

where $\boldsymbol{W}^{\text{temp}}_i \in \mathbb{R}^{1 \times 3d}$ is the learnable weight of the $i$-th expert, $\odot$ is element-wise multiply, and $g'_i$ represents the routing weight of the $i$-th expert. By this means, we can model user dynamic interest changes via two forms of time, interval and absolute time, to embed explicit temporal data into multi-modal learning.

Through the two-level hierarchical MoE, we can improve item key feature extraction by modality interactions and also introduce explicit temporal information to multi-modal learning for user dynamic interest modeling.

## 4.3 User Interest Learning and Prediction

To learn user evolving interest, we choose Transformer [29] to encode the information within behavior sequences effectively, which is widely applied in SR methods. Specifically, for the behavior sequence $S$ of the user $u$, we organize the outputs of all items from Temporal MoE as three types of sequence representations $S_m = \{\boldsymbol{x}^{\text{temp}}_{m,1}, \boldsymbol{x}^{\text{temp}}_{m,2} \ldots, \boldsymbol{x}^{\text{temp}}_{m,|S|}\}$, where $m \in \{id, txt, img\}$ represents the specific modality. Then, these sequences are fed into their corresponding Transformer models as follows:

$$S'_m = \text{Dropout}(\text{LayerNorm}(S_m)), \tag{17}$$

$$\boldsymbol{H}^m = \text{Transformer}_m(S'_m), \tag{18}$$

where $\boldsymbol{H}^m$ is the final hidden vector corresponding to the last position as user interest under modality $m$. Next, to predict the interaction probability between the user $u$ and the item $v$, we calculate the score on each modality and then sum up these scores as the final prediction result $\hat{y}_v$, which can be formulated as below:

$$\hat{y}_v = \boldsymbol{H}^{id} \cdot \boldsymbol{x}_{id} + \boldsymbol{H}^{txt} \cdot \boldsymbol{x}_{txt} + \boldsymbol{H}^{img} \cdot \boldsymbol{x}_{img}. \tag{19}$$

## 4.4 Multi-Task Learning

To alleviate information loss owing to data sparsity in multi-modal SR, we design the multi-task training strategy besides the main SR task to provide multi-faceted and informative supervision signals as complements. First, given a user behavior sequence $S$, the main target of SR is to predict which item would this user be interested in at the $(n+1)$-th time step. Through the proposed network above, we get the prediction score $\hat{y}_v$, which is the possibility that this user would interact with $v$ at the next time step. Then the cross-entropy loss function is utilized as the main training objective to measure the differences between our prediction and the ground truth $y_v$:

$$L_{\text{main}} = -\sum_{v \in \mathcal{V}} y_v \log(\hat{y}_v). \tag{20}$$

Next, considering that item category information is helpful for item feature understanding [42], we design the sequence-level category prediction task (CP) to provide category signals on multi-modal learning. To be specific, we take the categories of items as classification labels to supervise the model to distinguish which classes these items belong to based on all modality information. To generate signals more related to user sequence, we compute this task at the sequence level. For the item $v \in S$, we employ the output $\boldsymbol{x}_{\text{temp}}$ of Temporal MoE of $v$ to calculate its probability to belong to each class, which is formulated as:

$$\hat{y}^{CP} = \boldsymbol{W}_{CP}\boldsymbol{x}_{\text{temp}} + \boldsymbol{b}_{CP}, \tag{21}$$

where $W_{CP} \in \mathbb{R}^{3d \times |C|}$ and $b_{CP} \in \mathbb{R}^{|C|}$ are the learnable weight and bias for CP respectively, $C$ is the set of the categories. Let $\hat{y}^{CP}_c$ denote the likelihood that $v$ belongs to $c \in C$, then the binary cross-entropy loss is utilized to facilitate multi-label category information [8, 33]:

$$L_{CP}(v) = -\sum_{c \in C} y^{CP}_c \log\left(\hat{y}^{CP}_c\right) + \left(1 - y^{CP}_c\right) \log\left(1 - \hat{y}^{CP}_c\right), \tag{22}$$

where $y^{CP}_c$ is the ground truth. Then the loss of the CP task on $S$ is $L_{CP} = \sum_{v \in S} L_{CP}(v)$, which provides sequence-level category signals to enhance multi-modal learning.

To obtain supervision signals from user sequence context and user interests, we also design the objective of contrastive learning on ID (IDCL) to improve alignments between user sequence representations and the ground truth. Since the embeddings of text and images are frozen in the training process, we only utilize this task on ID. Specifically, within a training batch $B$, we organize the representations of users and their ground truth items on ID as $\{\langle \boldsymbol{H}^{id}_1, \boldsymbol{x}^{id}_1 \rangle, \langle \boldsymbol{H}^{id}_2, \boldsymbol{x}^{id}_2 \rangle, \ldots, \langle \boldsymbol{H}^{id}_{|B|}, \boldsymbol{x}^{id}_{|B|} \rangle\}$. We aim to make the vectors of user sequences and their ground truth be more similar, while forcing this user to be dissimilar with other items. Therefore, the loss of IDCL is formulated as:

$$L_{IDCL} = -\log \frac{\exp\left(\text{sim}\left(\boldsymbol{H}^{id}_i, \boldsymbol{x}^{id}_i\right)/\tau\right)}{\sum_{j=1}^{|B|} \exp\left(\text{sim}\left(\boldsymbol{H}^{id}_i, \boldsymbol{x}^{id}_j\right)/\tau\right)}, \tag{23}$$

where $sim(\cdot, \cdot)$ is the cosine similarity function, and $\tau$ denotes the temperature parameter.

Last, to deepen correlation learning between explicit temporal information and multi-modal data, we further propose a sequence-level augmentation contrastive learning method, called Placeholder Contrastive Learning (PCL). In particular, for the sequence $S = \{v_1, v_2, \ldots, v_{|S|}\}$, we randomly replace items with the Time Placeholder "[TP]" with the proportion $\beta$ to get the augmented sequence. Next, to integrate explicit temporal information to multi-modal data, for the representation sequence $S_m = \{\boldsymbol{x}^{\text{temp}}_{m,1}, \boldsymbol{x}^{\text{temp}}_{m,2}, \ldots, \boldsymbol{x}^{\text{temp}}_{m,|S|}\}$ on the modality $m$ after Temporal MoE, if the $i$-th item is replaced by "[TP]" in $S'$, then $\boldsymbol{x}^{\text{temp}}_{m,i}$ will be replaced with the operation below:

$$\boldsymbol{x}^{\text{temp}}_{m,i} \leftarrow \boldsymbol{W}_{P,m}[\boldsymbol{r}_{1,i}; \boldsymbol{r}_{2,i}] + \boldsymbol{b}_{P,m}, \tag{24}$$

where $\boldsymbol{r}_{1,i}$ and $\boldsymbol{r}_{2,i}$ are the $i$-th time embeddings from Temporal MoE, and $\boldsymbol{W}_{P,m} \in \mathbb{R}^{2d \times d}$ and $\boldsymbol{b}_{P,m} \in \mathbb{R}^d$ are trainable parameters. After that, the augmented sequence $S'_m$ is inputted into Transformer$_m$, and we can get the augmented representation $\boldsymbol{H}^{m,1}$ corresponding to $\boldsymbol{H}^m$. Subsequently, within a training batch $B$, we can obtain $\{\boldsymbol{H}^{m,1}_1, \boldsymbol{H}^{m,1}_2, \ldots, \boldsymbol{H}^{m,1}_{|B|}\}$ from $\{\boldsymbol{H}^{txt}_1, \boldsymbol{H}^m_2, \ldots, \boldsymbol{H}^m_{|B|}\}$, and we treat $\langle \boldsymbol{H}^m_i, \boldsymbol{H}^{m,1}_i \rangle$ as the positive pair, while $\{\langle \boldsymbol{H}^m_i, \boldsymbol{H}^{m,1}_j \rangle | i \neq j\}$ are regarded as negative pairs. We aim to enhance consistency between positive sequence pairs while make negative pairs less similar, thus

PCL is formulated as follows:

$$L_{PCL}^m = -\log \frac{\exp\left(\text{sim}\left(H_i^m, H_i^{m,1}\right)\right)}{\sum_{j=1}^{|B|} \exp\left(\text{sim}\left(H_i^m, H_j^{m,1}\right)\right)}. \tag{25}$$

Then the total loss of PCL is added up as:

$$L_{PCL} = (L_{PCL}^{txt} + L_{PCL}^{img})/2. \tag{26}$$

Finally, we sum up these auxiliary training tasks with the main recommendation task to enhance the learning quality of our model:

$$L = L_{main} + \lambda_1 L_{CP} + \lambda_2 L_{IDCL} + \lambda_3 L_{PCL}, \tag{27}$$

where $\lambda_1$, $\lambda_2$ and $\lambda_3$ are weight hyper-parameters for CP, IDCL and PCL respectively. By above three tasks, we generate multi-faceted supervision signals to mitigate information loss from data sparsity.

## 5 Experiment

In this section, we introduce details of the experimental setup and compare HM4SR with state-of-the-art SR baselines. Then the ablation study and the hyper-parameter study are conducted to display the impact of designed modules and hyper-parameters on model performance. We further discuss the effect of time encoding methods for Temporal MoE. Last, we present a case study to show how explicit temporal information influences recommendation results.

### 5.1 Experimental Setup

*5.1.1 Datasets.* Four public datasets are chosen from Amazon Review Datasets [12] to evaluate SR models, including "Toys and Games" (Toys), "Video Games" (Games), "Beauty" and "Home and Kitchen" (Home). For each dataset, duplicated interactions are removed, and interactions of each user are sorted by timestamps chronologically to build behavior sequences. Following previous studies [15, 19, 33], we filter out users and items that have fewer than five interactions to get the 5-core subset of each dataset. For text, we concatenate phrases from title, category and brand fields consistent with previous studies [1, 15, 30]. For images, we directly download the first image of each item based on provided product URLs. The statistics of the processed datasets are shown in Appendix A.1.

*5.1.2 Compared Methods.* To verify the effectiveness of our method, we select the following representative and competitive baselines for sequential recommendation from three categories:

- **Traditional Non-Time-Aware Methods**. GRU4Rec [13] and SASRec [19] adopt gated recurrent units and self-attention mechanisms respectively to learn user sequential interest. CORE [14] unifies the representation space of encoding and decoding steps to make prediction consistent. LRURec [38] designs linear recurrent units to improve encoding quality of long user sequences.
- **Traditional Time-Aware Methods.** TiSASRec [20] designs a time interval-aware self-attention mechanism. FEARec [9] conducts frequency-level sequence analysis with a ramp structure for information from the frequency domain. TiCoSeRec [5] devises five time interval-related sequence-level augmentation methods to obtain uniform sequences and applies contrastive learning.
- **Multi-Modal Methods.** NOVA [23] designs a non-invasive self-attention mechanism to exploit side information. DIF-SR [33]

devises the decoupled side information fusion module to mitigate rank bottleneck and improve the expressiveness of non-invasive attention matrices. UniSRec [15] transfers textual semantic information by MoE. MISSRec [30] designs Interest Discovery Module to grasp deep relations among items, modalities and preferences. M3SRec [1] uses modal-specific and cross-modal MoE to enhance modality learning in Transformers. IISAN [11] exploits decoupled parameter-efficient fine-tuning on modality foundation models to achieve both intra-modal and inter-modal adaption. TedRec [35] achieves sequence-level semantic fusion on text and ID by Fast Fourier Transform from the frequency domain.

*5.1.3 Evaluating Metrics.* The performances of the models are evaluated with Normalized Discounted Cumulative Gain (NDCG@K) and Mean Reciprocal Rank (MRR@K), where $K \in \{5, 10\}$. We adopt the *leave-one-out* evaluation strategy to conduct the experiments, which uses the last item and the penultimate item of each sequence as the test and validation item respectively, and the rest items for training. To calculate the metrics, the ranking scores are computed on the whole item set without sampling.

*5.1.4 Implementation Details.* We implement our method in Pytorch based on a widely used open-source library RecBole[1] [41]. We set the training batch size as 1024, and the hidden size of all methods is 64. The maximum length of each behavior sequence is limited to 50. For the encoder structure, the number of multi-head and the number of self-attention layers for the Transformer are both empirically set as 2. For the hyper-parameters, the expert number for Interactive MoE $k_1$ and Temporal MoE $k_2$ is selected from $\{2, 4, 6, 8, 12\}$, and $\mu$ is set as 100. $freq$ is empirically set as 10000. $\tau$ is searched within $\{0.1, 0.2, 0.3, 0.5, 0.7, 1.0\}$. The loss weight $\lambda_1$ is empirically set as 1.0, while $\lambda_2$ and $\lambda_3$ are tuned within [0.25, 1.5] stepping by 0.25 and $\{0.1, 0.3, 0.5, 0.7, 1.0, 1.5\}$ respectively. $\beta$ is chosen from [0.1, 0.5] stepping by 0.1. In addition, we use Adam optimization and the learning rate is 1e-3. We apply grid search to find the best settings for HM4SR and baselines. The details for baselines are shown in Appendix A.4. All experiments are conducted on a server with Intel(R) Xeon(R) Gold 6226R 16-Core CPUs and NVIDIA GeForce RTX 3090 GPUs.

### 5.2 Overall Comparison

Table 1 shows the evaluation results of HM4SR and other compared methods. We have the following findings. **(1)** Overall, HM4SR outperforms both traditional and multi-modal approaches. It achieves improvements ranging from 2.13%~22.9% compared to the best multi-modal method. Compared to the best time-aware traditional models, HM4SR gains the relative performance improvements ranging from 13.6%~64.1%. **(2)** Multi-modal methods outperform traditional models in most cases, indicating that modal information is important for richer user interest learning as the supplementation of ID modality. Yet HM4SR performs the best among them, showing the advancement of incorporating explicit temporal information to multi-modal modeling for capturing user preference changes. **(3)** Time-aware approaches (including TedRec) generally perform better than non-time-aware traditional models. This implies that contextual information held in temporal information is useful for

---

[1]https://github.com/RUCAIBox/RecBole

**Table 1: Overall performance comparison. Bold scores represent the highest results among all the methods, while the second highest scores are underlined. † denotes that the corresponding method is reproduced by ourselves.**

| Method | Toys | | | | Games | | | | Beauty | | | | Home | | | |
|---|---|---|---|---|---|---|---|---|---|---|---|---|---|---|---|---|
| | NDCG | | MRR | | NDCG | | MRR | | NDCG | | MRR | | NDCG | | MRR | |
| | @5 | @10 | @5 | @10 | @5 | @10 | @5 | @10 | @5 | @10 | @5 | @10 | @5 | @10 | @5 | @10 |
| GRU4Rec | 0.0236 | 0.0289 | 0.0201 | 0.0222 | 0.0385 | 0.0514 | 0.0311 | 0.0365 | 0.0271 | 0.0337 | 0.0223 | 0.0260 | 0.0067 | 0.0087 | 0.0056 | 0.0064 |
| SASRec | 0.0348 | 0.0411 | 0.0257 | 0.0295 | 0.0391 | 0.0546 | 0.0286 | 0.0349 | 0.0325 | 0.0415 | 0.0246 | 0.0283 | 0.0118 | 0.0148 | 0.0089 | 0.0100 |
| CORE | 0.0155 | 0.0273 | 0.0100 | 0.0148 | 0.0212 | 0.0360 | 0.0149 | 0.0210 | 0.0154 | 0.0266 | 0.0103 | 0.0150 | 0.0051 | 0.0080 | 0.0034 | 0.0046 |
| LRURec | 0.0362 | 0.0446 | 0.0278 | 0.0312 | 0.0410 | 0.0557 | 0.0315 | 0.0376 | 0.0323 | 0.0412 | 0.0250 | 0.0286 | 0.0112 | 0.0141 | 0.0084 | 0.0096 |
| TiSASRec | 0.0372 | 0.0467 | 0.0278 | 0.0317 | 0.0406 | 0.0568 | 0.0291 | 0.0357 | 0.0340 | 0.0432 | 0.0264 | 0.0302 | 0.0117 | 0.0146 | 0.0088 | 0.0100 |
| FEARec | 0.0354 | 0.0442 | 0.0266 | 0.0302 | 0.0411 | 0.0558 | 0.0306 | 0.0367 | 0.0342 | 0.0434 | 0.0265 | 0.0304 | 0.0121 | 0.0152 | 0.0092 | 0.0105 |
| TiCoSeRec | 0.0336 | 0.0404 | 0.0280 | 0.0311 | 0.0463 | 0.0589 | 0.0382 | 0.0434 | 0.0335 | 0.0413 | 0.0285 | 0.0317 | 0.0108 | 0.0134 | 0.0091 | 0.0102 |
| NOVA† | 0.0381 | 0.0478 | 0.0288 | 0.0328 | 0.0417 | 0.0576 | 0.0303 | 0.0368 | 0.0347 | 0.0442 | 0.0273 | 0.0312 | 0.0120 | 0.0147 | 0.0092 | 0.0102 |
| DIF-SR | 0.0359 | 0.0444 | 0.0284 | 0.0318 | 0.0416 | 0.0575 | 0.0311 | 0.0375 | 0.0340 | 0.0436 | 0.0262 | 0.0302 | 0.0092 | 0.0112 | 0.0078 | 0.0086 |
| UniSRec | 0.0276 | 0.0384 | 0.0194 | 0.0239 | 0.0386 | 0.0535 | 0.0291 | 0.0353 | 0.0287 | 0.0398 | 0.0212 | 0.0288 | 0.0121 | 0.0160 | 0.0092 | 0.0108 |
| MISSRec | 0.0323 | 0.0414 | 0.0235 | 0.0270 | 0.0398 | 0.0506 | 0.0312 | 0.0353 | 0.0315 | 0.0397 | 0.0240 | 0.0271 | 0.0140 | 0.0174 | 0.0107 | 0.0119 |
| M3SRec† | 0.0416 | 0.0486 | 0.0363 | 0.0391 | 0.0493 | 0.0643 | 0.0404 | 0.0464 | 0.0345 | 0.0428 | 0.0294 | 0.0328 | 0.0122 | 0.0144 | 0.0105 | 0.0115 |
| IISAN | 0.0395 | 0.0489 | 0.0354 | 0.0393 | 0.0525 | 0.0671 | 0.0432 | 0.0491 | 0.0372 | 0.0464 | 0.0309 | 0.0347 | 0.0152 | 0.0187 | 0.0129 | 0.0142 |
| TedRec | 0.0318 | 0.0397 | 0.0265 | 0.0297 | 0.0468 | 0.0604 | 0.0384 | 0.0439 | 0.0330 | 0.0419 | 0.0275 | 0.0311 | 0.0113 | 0.0140 | 0.0094 | 0.0105 |
| HM4SR | **0.0503** | **0.0573** | **0.0446** | **0.0475** | **0.0557** | **0.0688** | **0.0470** | **0.0524** | **0.0420** | **0.0493** | **0.0367** | **0.0396** | **0.0168** | **0.0191** | **0.0151** | **0.0160** |
| Improv. | +20.9% | +17.2% | +22.9% | +20.9% | +6.10% | +2.53% | +8.80% | +6.72% | +12.9% | +6.25% | +18.8% | +14.1% | +10.5% | +2.13% | +17.1% | +12.7% |

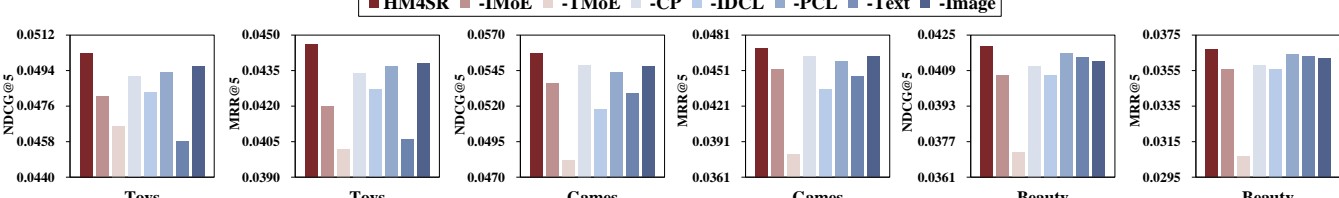

**Figure 3: Performance comparison of removing each designed component or each modality.**

user dynamic interest learning. However, HM4SR displays further improvement with the utilization of both explicit temporal information and multi-modal data. **(4)** HM4SR improves relatively less on Games Dataset compared to other three datasets. A possible reason is that the time distribution of user behaviors in Games is more unified, reducing the difficulty of user dynamic interest modeling. We present the time distributions of four datasets in Appendix A.5.

### 5.3 Ablation Study

To verify the effectiveness of each design for HM4SR, we compare our model with the following variants: **(1)** *-IMoE* removes the structure of Interactive MoE. **(2)** *-TMoE* removes Temporal MoE. Notably, $L_{PCL}$ is also removed simultaneously. **(3)** *-CP* removes the category prediction task. **(4)** *-IDCL* removes the objective of contrastive learning on ID. **(5)** *-PCL* removes the placeholder contrastive learning task. **(6)** *-Text* removes text modality. **(7)** *-Image* removes image modality. The evaluation results are demonstrated in Figure 3. All the components and modalities contribute to the performance of HM4SR. We can also observe that Temporal MoE generally achieves the highest improvement among all components, indicating the crucial importance of the incorporation of explicit temporal information for multi-modal modeling. PCL further amplifies this enhancement by deepening correlation learning between modality and time embeddings. Interactive MoE is useful for preference learning by improving item key information extraction. CP

and IDCL tasks also play a vital role in supervised signal generation. Notably, text information is usually more powerful than images since text portrays item features more directly.

### 5.4 Hyper-Parameter Study

In this part, we study the impact of important hyper-parameters, including the number of experts for Interactive MoE and Temporal MoE $k_1$ and $k_2$ respectively, the loss weights for IDCL and PCL $\lambda_2$ and $\lambda_3$, and the placeholder proportion $\beta$. When changing one hyper-parameter, we keep other hyper-parameters fixed to control variables. The results are illustrated in Figure 4. First, for the expert number of Interactive MoE and Temporal MoE $k_1$ and $k_2$, we can see that sufficient experts can significantly improve the performance. Whereas as the expert number further increases, the performance may drop slightly, probably due to overfitting. Second, two loss weight hyperparameters $\lambda_2$ and $\lambda_3$ control the strength of $L_{IDCL}$ and $L_{PCL}$. The results demonstrate that reasonably setting them can effectively enhance the preference learning ability of HM4SR. Making them either too high or too low will decrease the effect of two corresponding auxiliary training tasks. Last, $\beta$ is the proportion that items get replaced by temporal placeholders in PCL. It's observed that a too small $\beta$ may lead to insufficient relationship learning between explicit time information and modality, while a too large $\beta$ can result in generated positive sample fluctuating, both reducing the advancement of PCL.

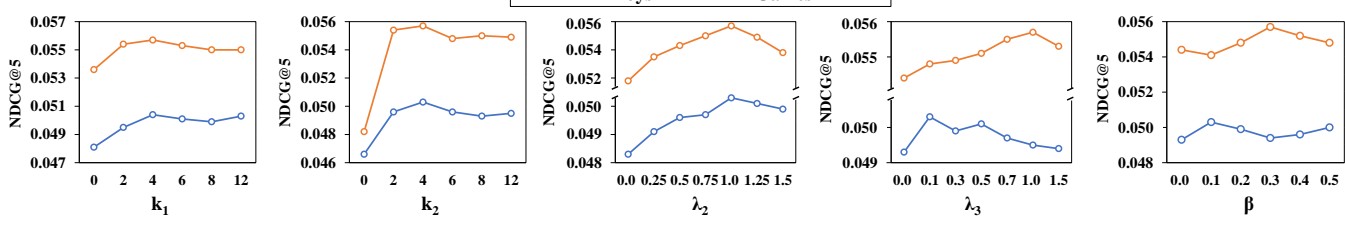

**Figure 4: Performance comparison of HM4SR w.r.t different hyper-parameters.**

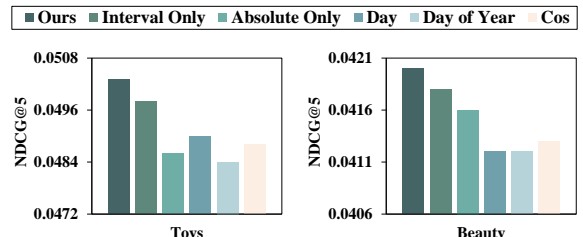

**Figure 5: Performance comparison on different time embeddings inputted to the gating router of Temporal MoE.**

## 5.5 Impact of Temporal Embeddings

For a deeper analysis of the impact of explicit temporal information on multi-modal SR, we change the types of time embeddings inputted to Temporal MoE as five variants: Besides **(1)** *Interval Only* only inputs $r_{1,i}$ and **(2)** *Absolute Only* only inputs $r_{2,i}$, we also try three widely-used types of time embeddings [3, 10, 28], including **(3)** *Day* embeds time based on the number of the day within the whole dataset, **(4)** *Day of Year* embeds time based on the number of the day within a year, and **(5)** *Cos* encodes the time interval $t$ into $[cos(w_1t + b_1), cos(w_2t + b_2), \cdots, cos(w_dt + b_d)]$. Figure 5 shows the result of all variants. Though all types of time embeddings can facilitate user preference learning, our time encoding method gets more advances, which is attributed to the combination of interval information and absolute timestamps. Meanwhile, we can also perceive our time interval embeddings are more flexible and useful than the interval embeddings calculated by *Cos*.

## 5.6 Case Study

We select a typical case from the test set of Toys to expound the effect of designed components, which is shown in Figure 6. We compare HM4SR with M3SRec which has the highest results of NDCG@5 and MRR@5 on Toys Dataset among the baselines. We can find that HM4SR successfully predicts the ground truth item, which is a sound-related toy. This can be attributed to three aspects: First, Interactive MoE fosters the extraction of item sound-related information from abundant multi-modal data. Second, Temporal MoE enhances user dynamic interest modeling and find that the user was more interested in toys with music and sound recently. Third, although this user sequence is short, supervision signals from the multi-task learning is helpful to deal with data sparsity and improve recommendation quality. By comparison, M3SRec analyses multi-modal data more generally and supposes that the user was keen on toys with animal figures. As a result, it puts toys with the same brand "Fisher Price" as the last item at a higher priority, and also recommends toys related to animal figures, both are less

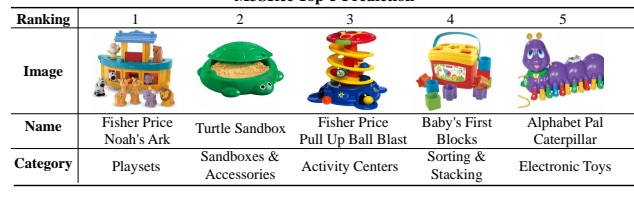

**Figure 6: A purchase sequence and the top-5 prediction results of HM4SR and M3SRec from Toys Dataset. For convenience, we only display item names and categories as text.**

precise results. These findings demonstrate the effectiveness of our design towards multi-modal SR.

## 6 Conclusion

In this paper, we proposed a multi-modal sequential recommendation method called HM4SR, which improved the prediction accuracy by extracting item key information and introducing explicit temporal information to multi-modal learning. Along this line, we designed a two-level hierarchical MoE. The first MoE, namely Interactive MoE, processed all modality information of items with experts. We also devised Temporal MoE as the second MoE, which encoded explicit temporal information to model user dynamic interests. To tackle information loss from data sparsity, a multi-task learning strategy was also invented. Besides the main SR task, the tasks of sequence-level category prediction (CP) and contrastive learning on ID (IDCL) are designed to provide fine-grained supervision signals. For deeper correlation learning between modalities and time, we further devised the Placeholder Contrastive Learning objective (PCL). Extensive experiments on public datasets demonstrated the effectiveness of the proposed method.

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

# A    Appendix

## A.1    Statistics of Datasets

We present the statistics of processed Toys, Games, Beauty and Home datasets in Table 2.

## A.2    Time Complexity Analysis

Given the batch size $B$, the maximum sequence length $n$ and the hidden size $d$, we compare the time complexity of HM4SR and M3SRec for inference. For HM4SR, time cost consists of following part: (1) To obtain initial item representations, HM4SR converts the embeddings of modalities into the same hidden size and add position embeddings, which cost $O(Bnd(d_{txt}+d_{img}+1))$. (2) Interactive MoE takes $O(Bnk_1(d^2+d))$ and $O(Bnk_1(d+1))$ for expert processing and the gating router respectively. (3) Experts of Temporal MoE spends $O(Bnk_2d)$, and the gating router takes $O(Bnk_2(d+1))$. (4) For user interest learning, the Transformer blocks cost $O(BL(n^2d+nd^2))$, where $L$ is the number of transformer layers. Overall, since the amounts of experts $k_1$ and $k_2$ are similar to $L$, the total time complexity of HM4SR can be summed up as $O(Bn(L(d^2+nd)+1)+Bnd(d_{txt}+d_{img}))$.

Similarly, the time complexity of M3SRec to inference includes the following parts: (1) The process for initial item representations is the same as HM4SR, which takes $O(Bnd(d_{txt}+d_{img}+1))$. (2) For the transformer blocks with modality-specific MoE, it spends $O(BL_1n^2d)$ for the self-attention mechanism, and the MoE costs $O(Bnk_3(d^2+d+1))$. Here, $L_1$ denotes the number of transformer blocks with modality-specific MoE, and $k_3$ is the expert amount of the modality-specific MoE. (3) For the transformer blocks with cross-modal MoEs, it takes $O(BL_2n^2d)$ and $O(Bnk_4(d^2+d+1))$ for the self-attention mechanism and cross-modal MoEs respectively, where $L_2$ represents the number of transformer blocks with cross-modal MoE, and $k_4$ is the expert amount of the cross-modal MoEs. Here we let $L=L_1+L_2$. (4) For the user representation fusion, it costs $O(Bd^2)$ for the attention mechanism. Overall, as the amounts of experts $k_3$ and $k_4$ are also similar to $L$, the general time complexity is rewritten as $O(Bn(L(d^2+nd)+1)+Bnd(d_{txt}+d_{img}))$. We can see that HM4SR and M3SRec have the same time complexity, yet our method outperforms the latter one significantly.

## A.3    Space Complexity Analysis

Let $\mathcal{U}$ and $\mathcal{V}$ denote the user set and the item set respectively, we compare the space complexity of HM4SR and M3SRec. For HM4SR,

**Table 2: Statistics of processed datasets.**

| Datasets | Toys | Games | Beauty | Home |
|---|---|---|---|---|
| # Users | 19,412 | 24,303 | 22,363 | 66,520 |
| # Items | 11,924 | 10,672 | 12,101 | 28,238 |
| # Actions | 167,597 | 231,780 | 198,502 | 551,682 |
| Avg. Actions/User | 14.58 | 9.53 | 8.88 | 8.29 |
| Avg. Actions/Item | 17.03 | 21.72 | 16.40 | 19.54 |
| # Sparsity | 99.93% | 99.91% | 99.93% | 99.97% |

space cost includes following parts: (1) For initial item representations, the embedding matrices for three modalities take $O(|\mathcal{V}|(d+d_{txt}+d_{img}))$. The two linear layers consumes $O((d_{txt}+d_{img}+1)d)$, while the position embeddings takes $O(nd)$. (2) Interactive MoE costs $O(k_1(d^2+d))$ and $O(k_1(d+1))$ for experts and gating router respectively. (3) The parameters for interval embeddings and absolute embeddings have $O((T^P+1)d)$ space complexity, where $T^P$ represents the total number of different values for the interval position $pos_i$. Experts of Temporal MoE spend $O(k_2d)$ and the gating router costs $O((k_2+d+1)d)$. (4) For user interest learning, the Transformer blocks cost $O(L(d^2+d))$. Overall, the amounts of experts $k_1$ and $k_2$ are similar to $L$. And since the intervals are processed by logarithmic function to get $pos_i$, $T^P$ is a relatively fixed value, thus we treat it as a constant. Therefore, the total space complexity of HM4SR is rewritten as $O(|\mathcal{V}|(d+d_{txt}+d_{img})+L(d^2+d)+(d_{txt}+d_{img})d)$.

Next, as for M3SRec, its space complexity contains the following parts: (1) The process for initial item representations is the same as HM4SR, which takes $O(|\mathcal{V}|(d+d_{txt}+d_{img})+(d_{txt}+d_{img}+1)d)$. (2) For the transformer blocks with modality-specific MoE, it spends $O(L_1d^2)$ for the self-attention mechanism, and the MoE costs $O(k_3(d^2+d+1))$. (3) For the transformer blocks with cross-modal MoE, it takes $O(L_2d^2)$ and $O(k_4(d^2+d+1))$ for the self-attention mechanism and cross-modal MoEs respectively. Here we let $L=L_1+L_2$. (4) For the user representation fusion, it costs $O(d^2)$ for the attention mechanism. Overall, as the amounts of experts $k_3$ and $k_4$ are also similar to $L$, the general space complexity of M3SRec is rewritten as $O(|\mathcal{V}|(d+d_{txt}+d_{img})+L(d^2+d)+(d_{txt}+d_{img})d)$. We can find that the space complexity of HM4SR is similar to M3SRec.

## A.4    Baseline Settings

We conduct our experiments based on Recbole[2]. The hidden size of baselines is fixed to 64. We apply grid search to find the best settings of the baselines with adjustable hyper-parameters. The codes of GRU4Rec, SASRec, CORE, TiSASRec and FEARec are directly taken from Recbole. We present further implementation details of other baselines as below:

- **LRURec**: We utilize the codes provided by the authors[3] and apply the settings in the original paper.
- **TiCoSeRec**: We take the official codes provided by the authors[4]. We follow the hyper-parameter searching scopes suggested by the authors.
- **NOVA**: We reproduce it based on the details from the original paper and the codes of DIF-SR. The sequence encoders are changed from BERT to Transformers for a fair comparison.

---

[2]https://github.com/RUCAIBox/RecBole
[3]https://github.com/yueqirex/LRURec
[4]https://github.com/kinggugu/ticoserec

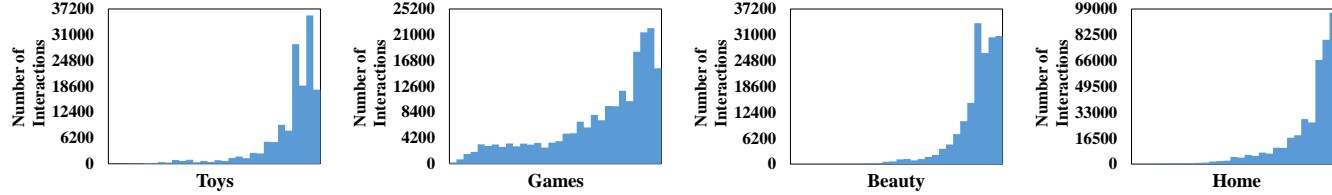

Figure 7: The time distributions of four datasets.

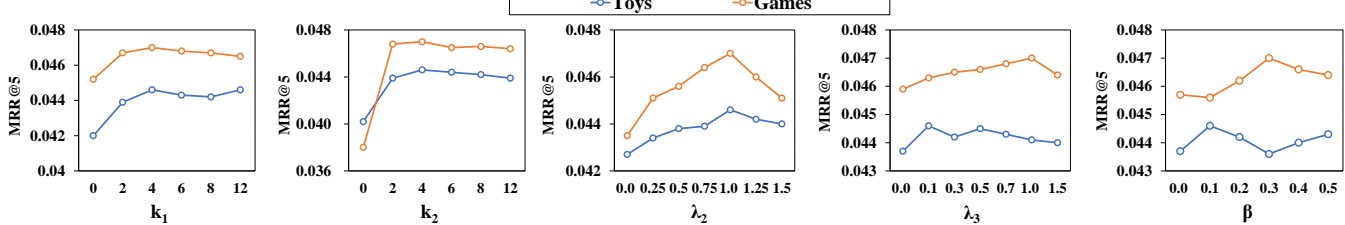

Figure 8: Performance comparison of HM4SR w.r.t different hyper-parameters on MRR@5.

- **DIF-SR**: The official codes[5] are utilized. The model settings are provided by the authors.
- **UniSRec**: We exploit the official codes[6]. The pre-training stage on cross-domain datasets is skipped for a fair comparison. We run it on the transductive setting, which utilizes both ID and text modalities.
- **MISSRec**: Its implementation is from the official codes[7]. The pre-training stage on cross-domain datasets is skipped for a fair comparison. We run it on the transductive setting, which utilizes ID and text and image modalities.
- **M3SRec**: We reproduce it and carefully choose its hyper-parameters according to descriptions from the original paper.
- **IISAN**: The implementation is provided by the authors[8]. The default settings of IISAN are applied.
- **TedRec**: We apply the original codes from the authors[9], and the settings are provided by the authors.

## A.5 Time Distributions of Datasets

To present the time distributions of a dataset, we obtain the maximum and minimum timestamps of interactions in the dataset, divide their time lag into 30 groups with equal intervals, and then count the number of interactions in each group whose timestamps belong to that group. The results of four datasets are illustrated in Figure 7. We can find that the time distribution of Games Dataset is significantly more unified than other datasets, which decreases the difficulty of user dynamic interest modeling.

## A.6 Addition Results of Hyper-Parameter Study

We show the results of parameter sensitivity on MRR@5 on Toys and Games datasets as the following Figure 8. They show a similar tendency with the results of NDCG@5 in Figure 4, which indicates

that inappropriate values of these hyper-parameters can hinder the improvement of HM4SR.

Received 20 February 2007; revised 12 March 2009; accepted 5 June 2009

---

[5]https://github.com/AIM-SE/DIF-SR
[6]https://github.com/rucaibox/unisrec
[7]https://github.com/gimpong/MM23-MISSRec
[8]https://github.com/GAIR-Lab/IISAN
[9]https://github.com/Sherry-XLL/TedRec

