# OpenReview forum: "Hierarchical Time-Aware Mixture of Experts for Multi-Modal Sequential Recommendation"
_ACM.org/TheWebConf/2025/Conference — WWW 2025 Poster_

### Official Review · Reviewer_J5La · 2024-11-29

**Novelty:** 5
**Technical Quality:** 5

**Review:**

This paper proposes a Hierarchical Time-Aware Mixture of Experts for Multi-Modal Sequential Recommendation (HM4SR). It aims to address the challenges in multi-modal sequential recommendation by extracting key information from multi-modal data and introducing explicit temporal information. The proposed method uses a two-level hierarchical MoE, including Interactive MoE and Temporal MoE, along with a multi-task learning strategy to enhance the model's performance. Experiments are conducted on four public datasets, showing the effectiveness of the proposed method.

pros：
1. The proposed HM4SR method effectively addresses the challenges in multi-modal sequential recommendation. By leveraging explicit temporal information and a hierarchical MoE structure, it outperforms both traditional and multi-modal baselines. For example, it achieves significant improvements (ranging from 2.13% - 22.9%) compared to the best multi-modal method and shows even greater advantages over traditional time-aware models.
2. The design of the two-level hierarchical MoE is innovative. Interactive MoE enables effective modality interaction learning by processing all modality information, enhancing item key feature extraction. Temporal MoE incorporates explicit temporal information, improving the model's ability to capture user dynamic interests. The ablation study clearly demonstrates the significant contribution of Temporal MoE to the overall performance, highlighting the importance of explicit temporal information in multi-modal modeling.
3. The multi-task learning strategy, including sequence-level category prediction (CP), contrastive learning on ID (IDCL), and placeholder contrastive learning (PCL), provides multi-faceted and informative training signals. This effectively mitigates information loss due to data sparsity and improves the model's preference learning ability. The ablation study also verifies the positive impact of these tasks on the model's performance.

cons:
1. The meaning conveyed by the figures is lacking. I could not get the sparseness of the user and item data represented in Figure 1. The relevance of the multi-task learning module and the overall framework is not represented in the overall framework Figure 2.
2. The effectiveness of the proposed method seems to vary across different datasets, especially on the Games Dataset where the improvement is relatively less. Although the paper attributes this to the more unified time distribution of user behaviors in Games, further investigation could be done to understand whether there are other factors contributing to this difference and how to address them. This could potentially lead to a more robust and universally applicable model.

**Questions:**

1. What does the paper mean by sparsity of user-item data, referring to items that have few interactions but are still representing the user's interests?
2. The paper mentions that textual information is usually more powerful than images because it characterizes items more directly. However, in some scenarios, users may choose based on the item's packaging or promotional images. How can the model be further enhanced to better handle and balance the contributions of different modalities, especially when the importance of the modalities may vary?

**Reviewer Confidence:**

4: The reviewer is certain that the evaluation is correct and very familiar with the relevant literature

**Scope:**

4: The work is relevant to the Web and to the track, and is of broad interest to the community

---

### Official Review · Reviewer_bbcu · 2024-12-02

**Novelty:** 3
**Technical Quality:** 4

**Review:**

This paper proposes a novel multi-modal sequential recommendation (SR) method, which incorporates two ideas: (1) hierarchical Mixture of Experts (MoE) to efficiently fuse modalities by exploiting temporal information, and (2) a multi-task learning strategy to leverage the data sparsity problem. This model outperforms other traditional non-time-aware and time-aware sequential recommendation methods and existing multi-modal sequential recommendation methods.

Pros:

1. The paper is well-structured, clearly presenting the objective, the method, and supporting experiments. It specifically explores various approaches to leveraging temporal information, a key factor in performance improvement, for the gate router to filter out interest-irrelevant modality features. The experiments support the analysis of how this temporal information enhances performance.

2. The proposed method shows the superior performance over the existing recommender systems.

Cons:

1. The experiments on whether multi-task learning addresses data sparsity are limited to simple ablation studies (removing each task) and case studies. Additionally, the sparsity levels of the datasets used are all similarly high (around 99%), which further limits the ability to differentiate the impact of each task. It would have been more informative to conduct experiments on datasets with varying sparsity levels or under scenarios like cold-start and warm-start settings.

2. There are no experiments comparing the proposed interactive MoE, which applies to all concatenated modalities, with the existing MoE approaches in previous multi-modal SR methods, which apply MoE to a single modality. Therefore, whether the proposed approach successfully achieves the objective of sufficiently addressing the lack of interaction between modalities when applying MoE is not adequately demonstrated.

3. It is already well-established that using both time intervals and explicit timestamps together tends to be more learnable. While the idea of using time information in MoE is promising, the novelty of how explicit time is utilized is lacking. Moreover, time information could have been more effectively used to independently learn the user's periodicity or seasonality. Instead, it was indirectly applied as a contrastive learning positive pair by randomly replacing items with time information, which raises questions about whether this approach fully leverages the potential of time information.

4. There is a minor typo: In line 285, encoder -> encode

**Questions:**

The other limitations are described above.

**Reviewer Confidence:**

3: The reviewer is confident but not certain that the evaluation is correct

**Scope:**

3: The work is somewhat relevant to the Web and to the track, and is of narrow interest to a sub-community

---

### Official Review · Reviewer_ThLb · 2024-12-02

**Novelty:** 4
**Technical Quality:** 5

**Review:**

HM4SR addresses two main issues, (1) interference from redundant, irrelevant information in multi-modal data and (2) the lack of explicit temporal signals for user interests. HM4SR consists of two Mixture of Experts (MoE) layers. the Interactive MoE for extracting relevant user interest information from multi-modal data, and the Temporal MoE for modeling dynamic user interests using explicit temporal embeddings. To handle data sparsity, the model uses three auxiliary tasks. Extensive experiments on four datasets show that HM4SR outperforms several state-of-the-art methods.

## Strong points
1. The motivation behind the proposed model is clear and well-justified, and the model design effectively addresses the identified challenges in multi-modal sequential recommendation.
2. The use of a hierarchical Mixture of Experts (MoE) is impressive: the Interactive MoE effectively extracts important user interests from multi-modal data, while the Temporal MoE integrates both types of temporal information, enhancing the model's ability to capture dynamic user interests.
3. The model illustration is well-drawn, making it easy to understand the overall architecture and the flow of information within the model.

## Weak points
1. PCL loss function (Line 557): The paper states that PCL learns the correlation between temporal information and multi-modal data, but it’s unclear how this loss term fulfills that role. It appears to function more as a contrastive learning loss, where augmented sequence representations of the same user are treated as positive pairs, and those from different sequences are negative pairs. This seems more related to sequence representation learning rather than temporal information integration.
2. Category prediction loss (L_CP): It's hard to understand why the L_CP loss helps improve performance, given that the model doesn't learn category embeddings directly. While the item category is used in the text embedding, the CP loss seems to rely on x_temp. This discrepancy requires further clarification.
3. IDCL loss and its relation to Eq. (20): The IDCL loss is described as encouraging the ID sequence representation and target ID embedding to become closer. However, it’s unclear whether this is performed in Eq. (20) as stated. If text and image embeddings are not trained, could it be that L_main from Eq. (19) implicitly includes the L_IDCL loss?

**Questions:**

### Here are some questions based on the weaknesses mentioned in the review:
1. How does the PCL loss effectively capture the correlation between temporal information and multi-modal data? It seems to focus more on contrastive learning for sequence representations rather than explicitly modeling temporal relationships. Could you clarify how this loss term contributes to temporal information integration?
2. The L_CP loss seems to rely on x_temp for category prediction, but the model doesn't learn category embeddings directly. Could you explain in more detail why the L_CP loss still improves performance despite not having category embeddings explicitly trained in the model?
3. The IDCL loss encourages the alignment of ID sequence representations and target ID embeddings, but it’s not clear how this aligns with Eq. (20). If the text and image embeddings are not being trained, does this mean L_main from Eq. (19) implicitly include the L_IDCL loss? Could you clarify this relationship?

### Other questions:
1. When combining different modalities, why is a simple concatenation of their representations used? Since the modalities may have different representation spaces, is there a need to align these spaces before combining them?
2. Would the model still work effectively if the two MoE components (Interactive MoE and Temporal MoE) were not performed hierarchically? Would the lack of a hierarchical structure impact the model’s performance?

**Reviewer Confidence:**

3: The reviewer is confident but not certain that the evaluation is correct

**Scope:**

4: The work is relevant to the Web and to the track, and is of broad interest to the community

---

### Official Review · Reviewer_qx7V · 2024-12-02

**Novelty:** 5
**Technical Quality:** 4

**Review:**

**Paper Summary**

This paper proposes a multi-modal sequential model with a two-level mixture of experts (MoE) and a multi-task learning strategy to address the following challenges. 1) Not all multi-modal data are relevant for making recommendations. 2) Explicit time intervals between item purchases are not considered. 3) Due to the sparse nature of data, modeling user interest through next-item prediction is challenging. Experimental results show that the proposed model outperforms existing models.

**Summary of Strengths**

S1. This paper is well-structured and provides a clear overview of the proposed model.

S2. By incorporating time-interval information into multi-modal data, this paper demonstrates significant performance improvements across four datasets.

S3. The ablation studies are carefully designed, thoroughly analyzing each component's contribution to the overall performance.

**Summary of Weaknesses**

W1. The paper does not compare its approach with recently published side-integrated sequential models like MSSR and ASIF. It is recommended to reference the following studies:

[1] Xiaolin Lin et al., “Multi-Sequence Attentive User Representation Learning for Side-information Integrated Sequential Recommendation,” WSDM 2024.

[2] Shuhan Wang et al., “Aligned Side Information Fusion Method for Sequential Recommendation,” WWW 2024.

W2. While the authors aim to tackle the issue of “hindrance of multi-modal redundant information” in existing multi-modal sequential recommendation systems using the Interactive MoE component, the paper does not include experiments showcasing this issue in real-world datasets. Additionally, it lacks qualitative or quantitative validation of the Interactive MoE’s ability to filter redundant data. Presenting MoE adapter weights in the Section 5.6 case study could highlight the model’s filtering capability.

W3. The proposed model is highly similar to the existing M3SRec study, except for adding the TMoE module. To strengthen the paper, the authors should evaluate TMoE’s effectiveness by integrating it with various baselines.

**Comments Suggestions and Typos**

C1. The role of placeholder contrastive learning in performance improvements is unclear. The method involves masking items in the sequence, pulling the masked and original items closer, and pushing other items apart, similar to the MLM approach in BERT [1]. However, the authors do not clearly explain how this technique integrates temporal information into multi-modal data, as claimed.

[1] Devlin, J., Chang, M., Lee, K., & Toutanova, K. (2019). BERT: Pre-training of Deep Bidirectional Transformers for Language Understanding. North American Chapter of the Association for Computational Linguistics.

C2. Although the authors provided code, it lacks training scripts, detailed instructions, and dataset preprocessing codes, making it challenging to reproduce the results. Including these elements would greatly improve the paper’s reproducibility.

C3. Figure 3 suggests that image data does not significantly enhance performance across datasets. The authors could discuss scenarios where incorporating image information might provide more substantial benefits for recommendation tasks.

C4. In Table 1, separating methods that use only ID and Text from those incorporating ID, Text, and Image would better illustrate the performance impact of different modalities.

**Questions:**

Q1. If the proposed model uses the same side information, such as attributes, brand, and category used in existing work, is it still better than existing models?

Q2. In the ablation study, the effect of the TM module is significant. I am wondering whether the TMoE module is particularly useful in real-world scenarios.

Q3. Regarding W2, I am wondering if the authors could provide a theoretical/empirical reason why placeholder contrastive learning helps the model learn temporal information?

**Reviewer Confidence:**

4: The reviewer is certain that the evaluation is correct and very familiar with the relevant literature

**Scope:**

3: The work is somewhat relevant to the Web and to the track, and is of narrow interest to a sub-community